# Reconfigurable infrared hyperbolic metasurfaces using phase change materials

T.G. Folland [1], A. Fali[2], S.T. White[3], J.R. Matson [4], S. Liu [5], N.A. Aghamiri[2], J.H. Edgar[5], R.F. Haglund Jr.[3,4], Y. Abate[2] & J.D. Caldwell [1]

Metasurfaces control light propagation at the nanoscale for applications in both free-space and surface-confined geometries. However, dynamically changing the properties of meta-surfaces can be a major challenge. Here we demonstrate a reconfigurable hyperbolic metasurface comprised of a heterostructure of isotopically enriched hexagonal boron nitride (hBN) in direct contact with the phase-change material (PCM) single-crystal vanadium dioxide ($VO_2$). Metallic and dielectric domains in $VO_2$ provide spatially localized changes in the local dielectric environment, enabling launching, reflection, and transmission of hyperbolic phonon polaritons (HPhPs) at the PCM domain boundaries, and tuning the wavelength of HPhPs propagating in hBN over these domains by a factor of 1.6. We show that this system supports in-plane HPhP refraction, thus providing a prototype for a class of planar refractive optics. This approach offers reconfigurable control of in-plane HPhP propagation and exemplifies a generalizable framework based on combining hyperbolic media and PCMs to design optical functionality.

[1] Department of Mechanical Engineering, Vanderbilt University, Nashville, TN 37212, USA. [2] Department of Physics and Astronomy, University of Georgia, Athens, GA 30602-2451, USA. [3] Department of Physics and Astronomy, Vanderbilt University, Nashville, TN 37212, USA. [4] Interdisciplinary Materials Science Program, Vanderbilt University, Nashville, TN 37212, USA. [5] Department of Chemical Engineering, Kansas State University, Manhattan, KS 66506, USA. Correspondence and requests for materials should be addressed to Y.A. (email: yabate@physast.uga.edu) or to J.D.C. (email: josh.caldwell@vanderbilt.edu)

Optical near and far fields can be manipulated by scattering light into the resonant modes of nanostructured materials, which collectively form optical metasurfaces[1–3]. Historically, metallic polaritonic elements[4] have been used, which are highly absorbing[5] and typically exhibit limited tuning due to the geometric dependence of optical resonances. True reconfigurability—that is, complete changing of the optical response—therefore becomes challenging, as it requires arbitrarily changing the shape of individual elements of the structure, dynamically controlling the local dielectric environment, or controlling the optical properties of the polaritonic material itself.

In this regard, phase-change materials (PCMs) offer an appealing approach to introducing true reconfigurability as they undergo significant changes in optical properties upon exposure to external stimuli[6,7]. Examples of PCMs are vanadium dioxide ($VO_2$)[8–11] and germanium antimony telluride (GeSbTe) glasses[6,12], which undergo dielectric to metallic phase transitions upon heating or pulsed-laser excitation. For $VO_2$, this is a volatile (non-latching) phase transition, whereas GeSbTe undergoes a non-volatile (latching) transition. By integrating PCMs and polaritonic materials, changes in optical properties induced by such a phase transition can provide the means to control the polariton dispersion by changing the local dielectric environment in which the evanescent polaritonic near-fields propagate. Thus, they can be exploited to realize reconfigurable metasurfaces[6,12–17]. However, one of the phases of PCMs is typically metallic and/or exhibits high optical losses. Consequently, in previous studies of surface-confined polaritons, such as surface plasmon or surface phonon polaritons, the propagation was restricted to spatial regions over the PCM where a low-loss dielectric phase was present[6,12]. This makes concepts such as nanophotonic waveguides, grating couplers and focusing elements extremely difficult to realize in PCM-surface-polariton-based systems, despite the opportunities available.

Here we exploit two key changes in approach that overcome these previous limitations. First, we significantly reduce losses in polariton propagation by using isotopically enriched hexagonal boron nitride[18,19] (hBN), a natural hyperbolic[20–24] medium that supports low-loss hyperbolic phonon polaritons (HPhPs). Second, by exploiting hyperbolic polaritons instead of the surface-confined variety[6,12–16], the polaritons remain sensitive to local changes in the dielectric function of the ambient environment[25], whereas the electromagnetic near-fields are strongly confined to the volume of the hyperbolic material[20,21,26]. This means that HPhPs can interact with spatially localized phase transitions of the PCM, yet do not suffer significant optical losses from this interaction, and thus should propagate over both metallic and dielectric phases. Crucially, we show this to be the case, and that the difference in the local dielectric environment between metallic and dielectric domains results in a large change in the HPhP wavelength in the hBN over each domain, which in turn results in the refraction of the polariton when transmitting across the PCM phase-domain boundaries. This means that the combination of hyperbolic media and PCMs employed here can be used to create refractive optical elements and waveguides[27], as well as components benefitting from full optical functionalities that to this point have been limited to far-field optics. We demonstrate such concepts using electromagnetic modeling, showing that PCM-HPhP heterostructures can be designed as optical resonators[20,28] and metasurfaces[29,30], as well as refractive near-field components, such as waveguides and lenses. This combination of PCMs with hyperbolic media opens a whole new toolset for near-field optical design and structuring. Significantly, for reversible PCM transitions, any of these designs can be fully reconfigured using either thermal changes or approaches based on laser writing. Finally, by exploiting the increasingly wide range

of different PCMs and hyperbolic materials and metamaterials, such as transition metal oxides[31], these effects can be realized over an extended range of frequencies.

## Results

**Near-field measurements of hyperbolic polaritons.** The prototype device (Fig. 1a, b) consists of a 24 nm-thick flake of [10]B-enriched hBN (~99% enriched[18,19]) transferred using low-contamination transfer techniques onto a single crystal of $VO_2$ grown on quartz. We use scattering-type scanning near-field optical microscopy (s-SNOM) to directly map and visualize the evanescent optical fields on the structure, corresponding to polaritonic waves of compressed wavelength $\lambda_p$, propagating primarily within the volume of the hBN slab (see Fig. 1a). In s-SNOM images, HPhPs can be observed in two ways: first, polaritons launched by the light scattered from the s-SNOM tip propagate to and reflect back from sample boundaries (e.g., a flake edge) creating interference fringes with spacing $\lambda_p/2$, which are scattered back to free space by the tip and detected[21,32,33]. Alternatively, polaritons can be directly launched from sample edges and propagate across the surface to interfere with the incident field at the tip, producing fringes with spacing $\lambda_p$[18,34]. Thus, in s-SNOM maps, a superposition of both so-called "tip-launched" and "edge-launched" fringes may be observed and are interpreted by considering the fringe spacing from individual waves ($\lambda_p/2$ vs. $\lambda_p$) and the direction of polariton propagation.

**Controlling hyperbolic polaritons using a PCM.** The presence of both tip- (wavelength $\lambda_p/2$, purple line in the $x$ direction) and edge-launched (wavelength $\lambda_p$, blue line in the $y$ direction) HPhPs can be readily observed in the hBN (Fig. 1c) slab transferred on top of the $VO_2$ single crystal. Here, this is visualized via the near-field amplitude s-SNOM image collected using a 1450 cm$^{-1}$ excitation laser at room temperature. The observation of both tip- and edge-launched modes in the $x$ direction, whereas only edge-launched modes being apparent along the $y$ direction derives from the properties of the boundaries in the heterostructure sample, namely the edges of the hBN and $VO_2$ crystals. As in previous experiments[18,21,34], the edge of the hBN crystal provides for both high reflection of tip-launched HPhPs as well as a sharp edge for direct initiation of edge-launched modes ($x$ direction). In contrast, the small size (440 nm thickness, 6.5 μm width) of the $VO_2$ crystal provides sufficient momentum to robustly scatter into HPhP modes at the $VO_2$ crystal edges (bottom/top edges in Fig. 1c)[18,34]. However, the interface between $VO_2$ and air at the crystal edge provides a significantly reduced reflection coefficient, which suppresses tip-launched waves, an effect observed in prior work[12,25]. A more detailed discussion of the occurrence of both tip- and/or edge-launched modes in the s-SNOM images is available in Supplementary Note 1 and Supplementary Fig 1.

Propagation of HPhPs is strongly influenced by the local dielectric environment[25,29], so we investigated the influence of the $VO_2$ phase transition by measuring the s-SNOM response of the sample as a function of temperature, traversing the full dielectric-to-metal transition from 60 °C to 80 °C[11]. The sample was heated in situ inside the s-SNOM microscope on a custom-built heating stage. Individual $VO_2$ domains are directly observed with s-SNOM due to the dielectric contrast between domains, with metallic (dielectric) $VO_2$ appearing as bright (dark) regions (Fig. 1d)[8–11]. As the device is heated further (Fig. 1e), the hBN-supported HPhPs are observed to propagate over both the metallic and dielectric domains of $VO_2$, for appreciable propagation distances in both regions. This contrasts with an earlier work focused on surface polaritons and PCMs, where the polaritons propagated for only a few cycles over the dielectric phase and

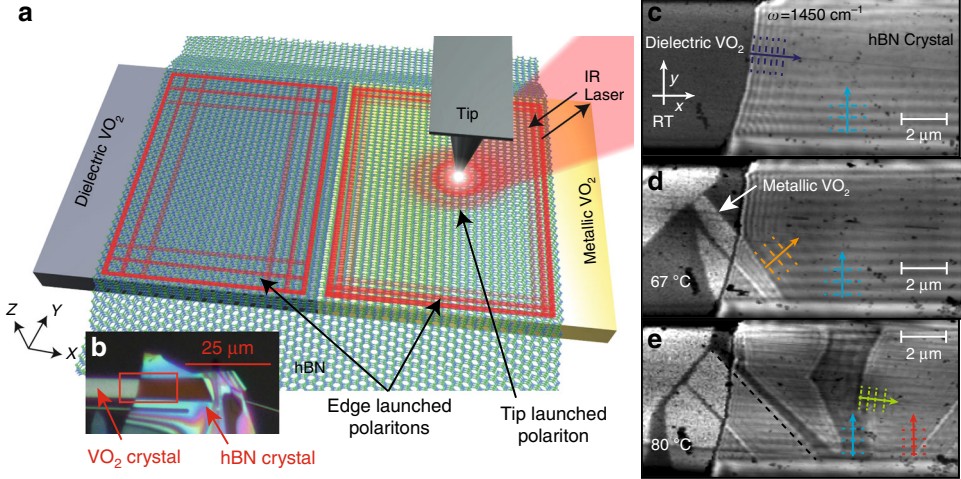

**Fig. 1** Actively reconfigurable hyperbolic metasurface device. **a** A device and experimental schematic, in which hBN has been transferred on top of a VO₂ single crystal and polaritons are imaged by the s-SNOM tip. **b** An optical microscope image of the heterostructure. **c–e** s-SNOM images of the optical near-field at 1450 cm⁻¹ (6.9 μm) at various temperatures, showing HPhPs propagating over both metallic and dielectric VO₂ domains. The complex patterns that form are the consequence of multiple interfering waves over the different domains. The arrows show the following: purple highlights tip-launched modes reflected from the hBN edge, whereas blue designates the HPhP propagating within the interior of the hBN from the edge of the dielectric VO₂ crystal (boundary with air, suspended hBN). The red arrow highlights the same propagation characteristics as the blue arrow, except for HPhPs propagating over the metallic VO₂ domains. Finally, the orange and green arrows designate HPhPs propagating within the hBN from the domain boundaries between the dielectric and metallic domains of the VO₂, with the orange (green) propagating over the dielectric (metallic) domains. In **e**, the black dashed line indicates the edge of the VO₂ metallic domain, extrapolated from the domain outside the hBN crystal

were entirely precluded from propagation over the metallic regions[12]. We attribute this difference to the volume confinement of the local electromagnetic near-fields of HPhPs supported within the low-loss hBN[18,20,21], which prevents the polaritonic fields from being absorbed by the lossy metallic phase of VO₂. After heating to high temperatures and performing these s-SNOM measurements, allowing the device to cool to room temperature resets the VO₂ crystal to its dielectric state, after which the sample can be reheated to get a different PCM domain pattern (see Supplementary Fig. 2). This allows us to reconfigure our device to study the propagation of HPhPs in a range of different geometries and at different frequencies within the same device. The large permittivity difference between metallic and insulating phases of VO₂ therefore presents an excellent platform to manipulate and control polariton propagation within hyperbolic materials.

When s-SNOM maps the evanescent fields of propagating HPhP waves in the presence of multiple interfaces, complex images result from the superposition of the waves launched, transmitted across and reflected by each domain boundary, crystal edge, and the s-SNOM tip. The simplest polaritons to identify are the modes launched from the edge of the VO₂ crystal, as they form straight fringes aligned parallel to the crystal edge. Due to the difference in local dielectric environment, these HPhPs possess different polariton wavelengths $\lambda_p$ above each domain. Here the HPhP mode launched by the VO₂ crystal edge over the dielectric (metallic) domain is highlighted by the light blue (red) arrow in Fig. 1c, d and demonstrate that the HPhP wavelength is modified from $\lambda/12.9$ to $\lambda/20.4$ by the PCM at 1450 cm⁻¹ between these domains, serving as the first report of the dispersion of HPhPs being tuned by a PCM. Propagation lengths (1/$e$) are approximately 2.83 μm (5.2 cycles) and 0.8 μm (2.5 cycles) in the dielectric and metallic phases at this frequency, respectively, which is comparable to propagation lengths in naturally abundant hBN (~3.1 and 2.5 μm at the same wavevectors, respectively)[21]. Furthermore, in Fig. 1d–e, s-SNOM images show that HPhPs are directly launched in the

hBN over the boundaries between the dielectric (orange arrow) and metallic (green arrow) domains, despite there being no appreciable change in the topography of the VO₂ crystal (Supplementary Note 2 and Supplementary Fig. 3). Although past work has shown that PCM domain boundaries can serve to launch polaritons[12], here they are launched and propagated over both phases, with a different wavelength over each, promising the potential for dynamically reconfiguring HPhP properties and propagation. Note that the VO₂ domains appear to change size when underneath the hBN (as seen by following the black dashed line in Fig. 1e). This arises from hyperlensing by the hBN[35,36], which acts to magnify light scattered into a hyperbolic medium and can give rise to spatial regions on the edge of a domain where the wavelength appears not to change (seen to the left of the blue arrow in Fig. 1e).

**Refraction of hyperbolic polaritons**. This heterostructure also enables the transmission of polaritons across the aforementioned domain boundaries. To simplify s-SNOM images and subsequent analysis, domain geometries with only a single boundary are required. As the positions of domain boundaries induced via thermal cycling of the VO₂ phase change are naturally quasi-random, we implemented multiple heating and cooling cycles (the same process as Supplementary Fig. 2) to achieve single dielectric-metal interfaces on the VO₂ crystal for study. Examples are shown in Fig. 2a, b (also Supplementary Note 3 and Supplementary Fig. 4). Such "reconfiguring" of the metasurface has been repeated more than eight times in our experiments, with no appreciable change in the dielectric properties of either of the two phases of VO₂ or the hBN flake, demonstrating the repeatability of this process.

Of particular interest is the polariton wave front that propagates away from the VO₂ crystal edge in the $y$ direction (purple dashed line with black arrows in Fig. 2a, b): it meets the domain boundary and distorts, propagating in a direction that is not normal either to the domain or crystal edge. This is a

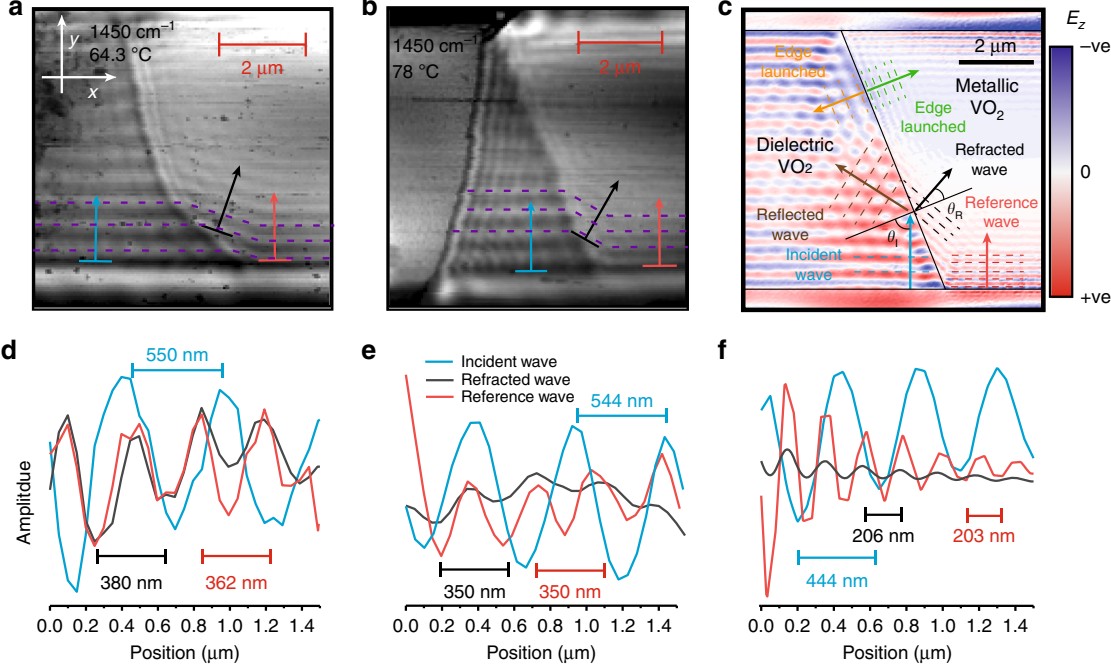

**Fig. 2** Hyperbolic polariton refraction within a hBN-VO$_2$ heterostructure. **a**, **b** Two s-SNOM maps of the near-field amplitude in the region of the domain boundary showing refraction. Purple dashed lines show the distorted phase front that propagates over the boundary. **c** An electromagnetic-field simulation of the geometry in **b**, showing reflected, refracted and edge-launched waves. **d–f** Line profiles from **a** to **c**, respectively, showing refraction of the wave

signature of planar polariton refraction as the wave changes direction due to the local change in dielectric environment. Although planar polariton refraction has been reported previously[37], this is the first direct observation of refraction for hyperbolic polaritons and the first evidence for such polariton refraction as a function of incident (transmitted) polariton angle.

If a hyperbolic polariton traverses the boundary between metallic and insulating VO$_2$ domains, the angle of propagation changes to conserve momentum in accordance with Snell's law:[38]

$$\frac{\sin(\theta_I)}{\sin(\theta_R)} = \frac{n_2}{n_1} \qquad (1)$$

where $n_1$ and $n_2$ are the indices of refraction in the first and second media, and $\theta_I$ and $\theta_R$ are the corresponding angles of incidence and refraction. To demonstrate that the experimentally measured images are due to refraction, we compare the results in Fig. 2b to a simplified electromagnetic simulation (Fig. 2c). In the simulation, we excite the structure with plane waves (45° incidence) and at the edges of the VO$_2$ crystal and polaritonic waves are launched that propagate across the surface, mimicking edge-launched polaritons. Note that we ignore the tip-sample interaction in these simulations. Instead, HPhPs excited at the edge of the VO$_2$ crystal (blue) propagate in the $y$ direction within the dielectric phase. When these HPhPs approach the angled dielectric-metallic domain boundary (black line), some of the wave will be reflected (brown) and some will be transmitted across the boundary (black) and refracted due to the mismatch in wavevectors for the HPhPs supported over the two PCM domains. The simulation also shows waves launched directly from the domain boundary (orange and green) in Fig. 1c, d. The refracted wave will not propagate normal to either the edge of the crystal or the domain boundary but will have the same polariton wavelength as the wave launched in the hBN by scattering of incident light from the metallic VO$_2$ crystal edge. This is indeed what is shown in our experiments by the corresponding line

profiles provided in Fig. 2d–f. However, the wave reflected by the metal-dielectric domain boundary is not observed experimentally due to interference with the edge-launched mode shown in light blue. Despite this, the good agreement between Fig. 2b and c shows clear evidence of HPhP refraction. Although in principle these effects should be observable also with a tip-launched waves in s-SNOM images, during our experiments however, we were unable to form a VO$_2$ domain boundary sufficiently close to the flake edge (seen in Fig. 1b) to study this effect.

**Quantifying polariton manipulation.** To quantify the change in the polariton wavevector and HPhP refraction induced by the VO$_2$ domains and to test the ability to reconfigure the metasurface, we systematically studied the polariton wavelength dependence on incident frequency and refracted angle in different domain geometries. In the first case, we systematically recorded s-SNOM images at several monochromatic incident laser frequencies in both metallic and dielectric domains, and subsequently extracted the polariton wavelength through Fourier analysis (see Supplementary Note 3 and Supplementary Fig. 5) of the s-SNOM maps, as has been reported previously[18,21,32,33]. The experimentally extracted polariton wavevector (symbols) agrees well with numerical calculations of the HPhP dispersion for thin hBN slabs on a substrate consisting of either the dielectric or metallic phase of VO$_2$ (Fig. 3a, b). In our assignment of the points in Fig. 3a, b, we consider both tip- and edge-launched modes, above both metallic and dielectric VO$_2$, which can be observed in Supplementary Fig. 5. Again, this dramatic change in wavevector between domains at the same incident frequency is attributable to the large change in dielectric constant in VO$_2$ between the two PCM states, which further compresses the polariton wavelength.

From the measured change in polariton wavelength, we calculated the ratio of the indices of refraction, $n_1/n_2$ to determine the expected angle of refraction for the HPhP waves from Eq. (1) and compared this with the refracted angle extracted from the s-SNOM images in Fig. 2 and Supplementary Fig. 4, to test the

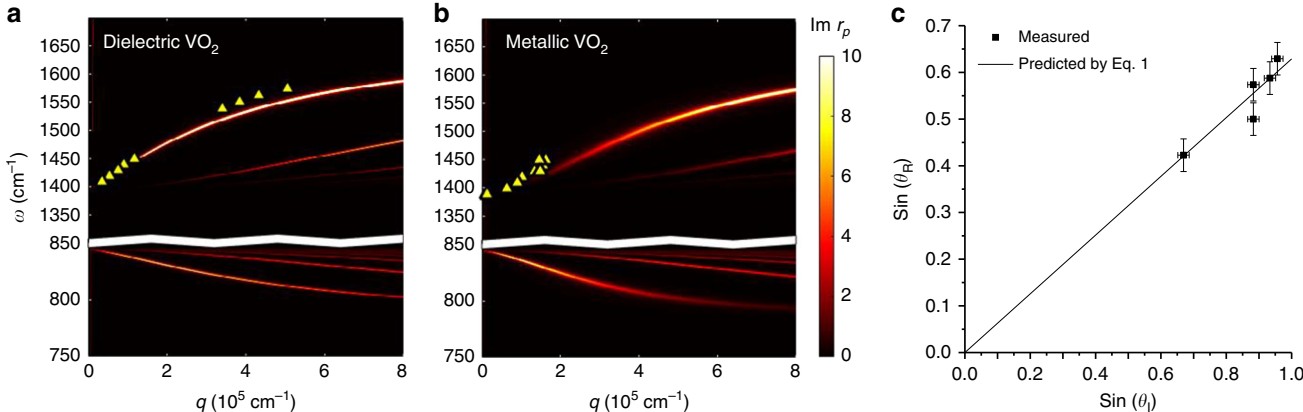

**Fig. 3** Hyperbolic polariton dispersion in hBN over both dielectric (**a**) and metallic (**b**) domains of $VO_2$ compared with numerical calculations. The horizontal white line shown in **a** and **b** indicates a break in the graph, between upper (1394–1650 cm$^{-1}$) and lower (785–845 cm$^{-1}$) Reststrahlen bands. From the measured dispersion, the angle of refracted waves at 1450 cm$^{-1}$can be computed for a given incident angle and compared against experimentally measured results in **c**. There has been no fitting in this result. The x and y error bars in **c** represent the SD of measurements of incident (±1°) and refracted (±2°) angles, respectively

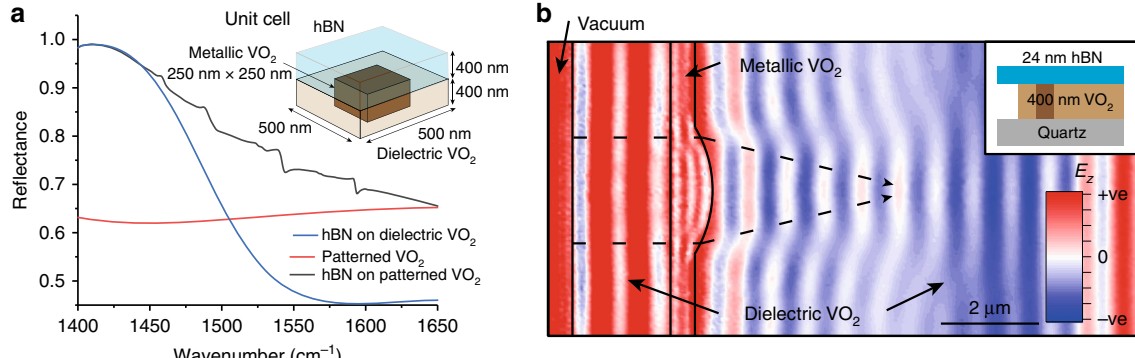

**Fig. 4** Schematic of refractive planar optics and reconfigurable resonators using phase-change materials. **a** A tunable polariton metasurface of hBN and $VO_2$, which consists of a continuous film of hBN 400 nm thick on top of 400 nm of $VO_2$. By patterning the $VO_2$ with metallic domains, we can excite a rewritable reflection profile, which cannot be generated from either of the materials alone. The pitch is 500 nm, with a particle length 250 nm. **b** A simulation of a refractive polariton lens, which uses a semi-circular domain of metallic $VO_2$ to launch polariton waves at 1418 cm$^{-1}$

adherence to Snell's law for HPhPs (Fig. 3c). This result is consistent with numerical simulations at a range of different angles and frequencies (see Supplementary Note 4 and Supplementary Fig. 6) confirming that Snell's law holds for HPhPs propagating across domain boundaries. Systematic investigation of polariton propagation and refraction at multiple angles was not possible in prior work[37] and thus the results presented here demonstrate that the tools and concepts of refractive optics are applicable in near-field optical designs as well. Indeed, the repeatable nature of both the change in polariton wavelength and Snell's law demonstrates that this platform can steer polariton propagation by proper design of the local dielectric environment.

**Towards refractive near-field optics.** The ability to control HPhPs propagating across phase-domain boundaries opens several possibilities for engineering lithography-free metasurfaces and near-field optics. As an example motivated by prior work[12], we investigated the possibility for creating rewritable nano-resonators using this technique, where a periodic array of metallic square domains is created inside the $VO_2$ crystal underneath the hBN (inset Fig. 4a). In Fig. 4a, we show the numerically calculated reflection spectrum from a hBN crystal on top of a dielectric $VO_2$ (blue curve), $VO_2$ patterned with metallic and dielectric domains (red curve), and hBN on top of such a patterned $VO_2$ structure

(black curve). In the simulated spectra for the hBN on top of patterned $VO_2$, there are peaks corresponding to a series of HPhP modes. Although these peaks are relatively small in amplitude (as this geometry has not been optimized for an intense resonant response), these modes can be tuned in frequency by changing the width and periodicity of the metallic domain (see Supplementary Note 5 and Supplementary Fig. 7 for a complete discussion). Thus, in principle, by controlling the size and shape of the metallic domain, one can realize a resonant response that previously was only observed in nanofabricated structures of hBN[20,39–41]. This implies that such resonators can be realized without the additional induced losses that are incurred with most nanofabrication approaches[42–44]. Such resonators could also be achieved experimentally by doping to change the local phase transition properties of $VO_2$[45].

Refraction of HPhPs across boundaries also enlarges the toolbox for near-field optics to include those of conventional refractive systems, such as in-plane lenses, whereby polaritons are focused to a point via refraction. A simulation of such a lens is shown in Fig. 4b, where HPhPs are launched into hBN at the left crystal edge and propagate inward to a region over a hemispherical $VO_2$ metallic domain, after which they are focused to a spot in the area over the dielectric $VO_2$. Here, the combination of hyperbolic media and PCMs is critical, because

for conventional surface polaritons, the high losses of the PCM metallic state would preclude polariton propagation and thus the polariton refraction required to induce focusing. Although experimentally we demonstrate the principle of this reconfigurable nano-optics platform using heterostructures comprising thin slabs of hBN on $VO_2$ single crystals, this approach can readily be generalized to other materials. To demonstrate this, we have simulated a nanophotonic waveguide using both $VO_2$ and $GeSbTe$[12] as the underlying PCMs (see Supplementary Note 6 and Supplementary Fig. 8). The non-volatile nature of the phase change in $GeSbTe$[12], where both states of the PCM are stable at room temperature, offers significant benefits for laser-writing-based approaches aimed at realizing complicated nanophotonic architectures. Although the device we present here is a conceptual prototype, our system could be realized in practice and scaled by using $VO_2$ or $GeSbTe$ films grown by sputtering and boron nitride grown by metal organic chemical vapor deposition[46]. There remain numerous material challenges—such as the growth of high-quality, large-area hBN—in realizing such a system, but this provides a route to achieving scalable reconfigurable devices.

## Discussion

We have experimentally demonstrated that the dispersion of HPhPs can be controlled using the permittivity changes inherent in the different phases of PCMs. This enables the direct launching, reflection, transmission, and refraction of HPhP waves at the domain boundaries between the different phases of the PCM, due to the large change in HPhP wavelength (here, by a factor of 1.6) that occurs for modes propagating in the hBN over each of these domains. Thermal cycling of the hBN-$VO_2$ heterostructure creates a range of domain-boundary geometries in the PCM, enabling the demonstration of various near-field phenomena. By inducing well-defined domain structures, it will be possible to design reconfigurable HPhP resonators and refractive optics in a planar, compact format at dimensions far below the diffraction limit. Beyond the implications for integrated nanophotonics, reconfigurable HPhP resonators could be used to match resonant frequencies to local molecular vibrational modes for the realization of dynamic surface-enhanced infrared absorption (SEIRA) spectroscopy[41]. Although in our case we have experimentally demonstrated these concepts using hBN on $VO_2$, using different combinations of PCMs (such as GeSbTe) and other hyperbolic materials (such as transition metal oxides[31]) could see expanded applications over a wide frequency range. Ultimately, we anticipate that the combination of low-loss, hyperbolic materials, and latchable PCMs will result in applications in lithography-free design and fabrication of optical and optoelectronic devices, whereas volatile PCMs could be used for dynamic modulation of photonic structures.

## Methods

**Device fabrication**. $VO_2$ single crystals were grown by physical vapor transport in a quartz tube furnace at 810 °C under 1.7 Torr Ar gas at a flow rate of 25 s.c.c.m. Vanadium pentoxide ($V_2O_5$) powder (~0.3 g, Sigma Aldrich 221899) was placed in a quartz boat (10 × 1 × 1 cm) upstream of the desired substrates and heated for 1 h. Evaporated $V_2O_5$ was reduced to $VO_2$ in this process and deposited on quartz (0001) substrates. Representative crystals from each sample were investigated using Raman spectroscopy to identify the $VO_2$ phase and optical microscopy to verify the thermal phase transition. Smaller, loose crystals located on the substrate surface were removed by adhesion to a heated (60 °C) layer of PMMA firmly brought into contact with the sample and subsequently retracted.

The isotopically enriched hBN crystals were grown from high-purity elemental $^{10}B$ (99.22 at%) powder by using the metal-flux method. A Ni-Cr-B powder mixture at respectively 48 wt%, 48 wt%, and 4 wt% was loaded into an alumina crucible and placed in a single-zone furnace. The furnace was evacuated and then filled with $N_2$ and forming gas (5% hydrogen in balance argon) to a constant pressure of 850 Torr. During the reaction process, the $N_2$ and forming gases continuously flowed through the system with rates of 125 s.c.c.m. and 25 s.c.c.m., respectively. All the nitrogen in the hBN crystal originated from the flowing $N_2$ gas.

The forming gas was used to minimize oxygen and carbon impurities in the hBN crystal. After a dwell time of 24 h at 1550 °C, the hBN crystals were precipitated onto the metal surface by cooling at a rate of 1 °C/h to 1500 °C, and then the system was quickly quenched to room temperature. Bulk crystals were exfoliated from the metal surface using thermal release tape. Crystals were subsequently mechanically exfoliated onto a PMMA/PMGI (polymethylglutarimide) polymer bilayer on silicon. Flakes were then transferred from the polymer substrate onto $VO_2$ single crystals using a semi-dry technique and the polymer membrane was removed using acetone and isopropyl alcohol.

**Numerical simulations**. Numerical simulations were conducted in CST Studio Suite 2017 using the frequency domain solver with plane waves incident at 45° and Floquet boundary conditions. In these simulations, polariton modes were only launched by scattering from edges in the simulation and field profiles were extracted using frequency monitors. All results used thicknesses consistent with that measured in topographic maps of the samples. Dielectric functions were taken from ref. [18] for isotopically enriched hBN, from ref. [47] for $VO_2$, and from ref. [48] for GeSbTe.

**sSNOM measurements**. Near-field nano-imaging experiments were carried out in a commercial (www.neaspec.com) s-SNOM based around a tapping-mode atomic force microscope. A metal-coated Si-tip of apex radius $R \approx 20$ nm that oscillates at a frequency of $\Omega \approx 280$ kHz and tapping amplitude of about 100 nm is illuminated by monochromatic quantum cascade laser laser beam at a wavelength $\lambda = 6.9$ μm and at an angle 45° to the sample surface. Scattered light launches hBN HPhPs in the device and the tip then re-scatters light (described more completely in the main text) for detection in the far-field. Background signals are efficiently suppressed by demodulating the detector signal at the second harmonic of the tip oscillation frequency and employing pseudo-heterodyne interferometric detection.

## Dataset availability statement

The datasets generated during and/or analyzed during the current study are available from the corresponding authors upon reasonable request.

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

## Acknowledgements

We thank Professor Misha Fogler for providing a script to calculate the dispersion of HPhPs. T.G.F. and S.T.W. thank the staff of the Vanderbilt Institute for Nanoscience (VINSE) for technical support during fabrication and Kiril Bolotin for preliminary design of the 2D transfer tool used. Support for the $^{10}$B-enriched hBN crystal growth was provided by the National Science Foundation, grant number CMMI 1538127. Y.A. and N.A. gratefully acknowledge support provided by the Air Force Office of Scientific Research (AFOSR) grant number FA9559-16-1- 0172. The work of A.F. is supported by the National Science Foundation grant 1553251.

## Author contributions

R.F.H, Y.A., and J.D.C. conceived and guided the experiments. S.T.W. grew the VO$_2$ crystals and identified the phase domains. S.L. and J.H.E. grew the hBN crystals. T.G.F. and S.T.W. fabricated the hBN-VO$_2$ heterostructure. A.F. and N.A. performed s-SNOM mapping experiments of the sample at various temperatures and incident frequencies. T. G.F. advised on experimental questions, developed the electromagnetic models, and analyzed s-SNOM data to show the presence of refraction. T.G.F. also conducted electromagnetic simulations of resonators, lenses, and waveguides. S.T.W. and J.R.M. analyzed s-SNOM data and calculated the dispersion curves. All authors contributed to writing the manuscript.

## Additional information

**Competing interests:** The authors declare no competing interests.

