## [Peer Review File · Nature Communications]

Reviewers' Comments:

Reviewer #1:

Remarks to the Author:

The manuscript is aimed at demonstrating a reconfigurable hyperbolic metasurface comprised of a hetero-structure of hexagonal boron nitride (hBN) in direct contact with the phase-change material single crystal vanadium dioxide (VO₂). The reported metasurface is supposed to overcome limitations caused by geometrically fixed structures, and to use materials with low propagation losses. The thermal treatment of the VO₂ layer makes metal-dielectric domains. This enables the direct launching, reflection, transmission and refraction of surface-polariton waves at the domain boundaries and propagating along the hBN/VO₂ interface.

That is an interesting heterostructure (hBN\VO₂) indeed. A number of cited in the manuscript papers, which are published over last 5 years regarding hBN and VO₂, confirms that. Note, that all properties of the surface polaritons in hBN, including sensitivity to the environment were reported in those papers. Majority of the results in the manuscript are referred to the previous works. In my opinion, the novelty of the results is limited to the minor further development relative to those papers.

Also, the conclusions are a little bit oversell the results. It is seen from the scans that the VO₂ domain boundaries create all possible optical responses. If we look at the second out of two Fig. 1, the scans in a) and b) show some patterns only at the corner of the domain, which are quickly disappear then. Thus, it is hard to claim low losses, long propagation distance of the refracted or diffracted polaritons. It is hard to call this approach as a real control of the surface polariton's propagation. Finally, the demonstrated realization of the hetero-structure is too far from integrated nanophotonics applications.

Summarizing, the manuscript is not suitable for publication in Nature Communications.

Reviewer #2:

Remarks to the Author:

The authors demonstrate the control of low-loss hyperbolic phonon polariton in hBN with the help of switching the phase of VO₂ that they use as a substrate. They further demonstrate refraction of the polaritons. Altogether they present a set of experiments that demonstrate the feasibility reconfigurable phonon polariton nanooptics, which could allow, for example, the development of tiny circuits (beam steering, waveguiding, cavities), which could enable novel mid-infrared sensing or signal processing application. The paper is for these reasons very interesting and attractive for the broad readership of Nature Communications. Before publication, the authors should elaborate a bit more some details and address the following points:

- 1) Numbering of figures needs to be corrected. E.g. there are two Fig. 1.
- 2) Why do we see either (exclusively?) tip- or edge-launched polaritons in Fig. 1b? What determines whether we see tip- or edge-launched polaritons?
- 3) In Fig. 1d one can see smaller fringe spacing on the metallic VO₂. How can the authors exclude that we do not see for some reason the tip-launched polaritons? The ratio in wavelength between metallic and dielectric VO₂ is close to a factor 2.
- 4) Fig. 1d: The second dark fringe (from the edge where the blue arrow is located) is at the same position on both the dielectric and the metallic VO₂ left of the blue arrow. Why does this metallic VO₂ not modify the polariton wavelength?
- 4) In the first paragraph of results the authors write that the polaritons launched from sample edges interfere with the tip. I don't understand what is meant. The polariton field at the position of the tip interferes with the incident field at the position of the tip.

5) It would be nice if the authors could show that the polariton wavelength at one specific location on the hBN becomes larger (the original one) when switching back from metallic to dielectric VO₂, in order to demonstrate reconfigurable control.

6) The authors refer in main text to Fig. S2 regarding multiple heating/cooling cycles. I don't understand what is shown in Fig. S2 in this regard. It is too little described.

7) The fringes associated with the refracted polaritons (marked by the black arrows) are not very well seen in the images. I suggest the authors to play with contrast to improve the fringe visibility and eventually show a zoom in to the region of interest. The dashed lines additionally mask the fringes.

8) I assume the numerical simulation Fig. 2c (respectively Fig. 1 according to Figure caption) does not include the tip. The authors should clarify that.

9) The fringes related to the refraction that we see in Fig. 2a,b seem to be all edge launched. Why are there no tip-launched polaritons seen (I see them only on the left hBN edge in Fig. 2b).

10) In section "Quantifying Polariton Manipulation" the authors mention a pump-laser. I don't understand why and what is pumped. I think the authors simply mean the laser used for imaging. In this case, the word "pump-laser" is misleading.

11) What is the white horizontal line in Figure 2 (respectively 3) and why is it slightly oscillating?

12) Fig. 3a (respectively 4a) I don't understand well. What does the inset show? The electric field distribution? What does it reveal?

In the reflectance graph one can see three spectra for different width L of the metal. The spectra change as a whole (why?). The authors state that via width L the peaks corresponding to a series of HPhP modes can be tuned (which indeed one can expect), however, the peaks (why so small?) marked by red arrow do not seem to change with L. The authors need to clarify this issue and elaborate the resonances and their tuning more detail.

13) The authors describe their work with the word "reconfigurable". I don't have clear how the metasurface is reconfigurable. This aspect is too little described and not demonstrated. The authors should elaborate this aspect. Particularly, they should describe in more detail how the sample preparation and experiment was done. E.g. was the VO₂ heated before or after the flake was deposited? At which temperature was imaging performed?

14) Generally, the authors could improve a bit the presentation of results: presentation of figures (labeling, font sizes (e.g. in S3) etc), description of figures in more detail, etc.

Reviewer #1 (Remarks to the Author):

That is an interesting heterostructure (hBN\VO2) indeed. A number of cited in the manuscript papers, which are published over last 5 years regarding hBN and VO2, confirms that. Note, that all properties of the surface polaritons in hBN, including sensitivity to the environment were reported in those papers. Majority of the results in the manuscript are referred to the previous works. In my opinion, the novelty of the results is limited to the minor further development relative to those papers.

Where there has certainly been work investigating VO₂ and hBN individually, there is not a single report that we could find discussing a heterostructure comprising both hBN and VO₂ or any other phase change material (PCM) for that matter. Further, while there is a report of surface polaritons integrated with PCMs (Ref. 12), here we are discussing hyperbolic polaritons, which are volume-confined. This distinction is of critical importance as we discuss in great detail in the manuscript. For example, in paragraph 3 of the original manuscript we stated "...by implementing *hyperbolic* polaritons instead of the surface-confined variety^{6,12-16}, the polaritons remain sensitive to local changes in the dielectric function of the ambient environment²², but the electromagnetic near-fields are strongly confined to the volume of the hyperbolic material¹⁹⁻²¹." This allows the unparalleled control of polaritonic fields, which is the main novelty of our paper. Thus, while we are unclear as to which citations the referee is referring to that he/she believes limits the novelty of our work, below we have provided a chart incorporating 10 references we cited and what they presented and how our work is distinct and novel. In addition, we have provided at the end of this response a more detailed discussion for each of these works, what the authors demonstrated and how this is different from our results presented here.

Reference number	8	12	13	14	15	16	17	25	29	30
Central points of this paper										
Hexagonal boron nitride (hBN)								X	X	X
Hyperbolic volume phonon polariton										X
Phase-change material (VO ₂ or GST)	X	X	X	X	X	X	X			
Polariton supported over both phases of PCM										
Reversibility and reconfigurability		X								
Polariton refraction										
Long polariton propagation length										
Central points of cited background papers										
Localized surface plasmon resonance	X									
Switching surface phonon polariton		X								
Switching and lensing on plasmonic metasurface			X							
Reconfigurable all-dielectric metasurface				X						

PCM switching on plasmonic nanorod arrays					X					
Ultrathin perfect absorber using VO ₂						X				
Elliptical to hyperbolic response in VO ₂ /TiO ₂							X			
Hyperbolic phonon polaritons over SiO ₂ and Au								X		
Hyperbolic surface phonon polaritons in hBN									X	
IR hyperbolic metasurface in structured hBN										X

As can be seen from this list, none of these prior references refer to a hyperbolic/PCM structure or metasurface, none demonstrate in-plane control of polariton propagation in a reconfigurable manner and none discuss direct observation of refraction, a fundamental property of polaritons, and the ability to apply this for realizing a broad array of advanced nanophotonic designs. Thus, it is unclear to us which references the reviewer is referring to that he/she believes indicates that the ‘majority of the results in the manuscript are referred to [in] the previous works’. We believe we have clearly shown that our results are indeed distinct from any of these prior efforts, and offer new physical insights, as well as provide the potential for novel applications and therefore believe our results meet the threshold for publication in *Nature Communications*. However, to make these points more clear, we have made additional clarifying remarks throughout the manuscript. In addition, to avoid misunderstandings and misrepresentations of our work, we have modified the first few sentences of the abstract to make sure the main findings of our work are clear from the start.

Also, the conclusions are a little bit oversell the results.

We reread our conclusion very carefully and we do not see any exaggerated claims. To make sure this is so, we revisited each sentence in our concluding paragraph and re-evaluate them here.

“In conclusion, we have for the first time experimentally demonstrated that the dispersion of HPhPs can be controlled using the permittivity changes inherent in the different phases of PCMs.”

- *Experimental control of both the wavelength and dispersion (the wavelength and energy) of the polariton is the heart of the paper, as demonstrated in Fig. 1b-d.*

“This enables the direct launching, reflection, transmission and refraction of HPhP waves at the domain boundaries between the phases of the PCM, due to the large change in HPhP wavelength (here a factor x1.6) that occurs for modes propagating in the hBN over each of these domains.”

- *The different domains of VO₂ enable manipulation of HPhP (Fig. 1b-d, Fig. 2, Fig. 3), the factor x1.6 is measured from experimental data.*

“Thermal cycling of the hBN-VO₂ heterostructure creates a range of PCM domain boundary geometries, enabling the demonstration of various near-field phenomena.”

- *Again, same as above, this sentence discusses the observation of the experimental results presented in Figs. 1-4.*

“By inducing well-defined domain structures, it will be possible to design reconfigurable HPhP resonators and refractive optics in a planar format at deep subdiffraction-limited dimensions.”

- *Our work demonstrates that nanostructured domains whose permittivity can be actively controlled can be prepared as heterostructures with hBN allowing reconfigurable HPhP behavior. We anticipate this ability could lead to potential applications such as tunable, reconfigurable resonators and refractive optics below the diffraction limit.*

“Beyond the implications for integrated nanophotonics, reconfigurable HPhP resonators could be used to match resonant frequencies to local molecular vibrational modes for the realization of dynamic surface-enhanced infrared absorption (SEIRA)³⁸ spectroscopy.”

- *We have demonstrated the polariton wavelength change actively using VO₂. Such a reconfigurable change of resonator response could allow enhanced spectroscopy of materials, with the reconfigurable nature providing the opportunity to tune the resonance between different molecular vibrational frequencies. This is especially pertinent in light of the work by Autore et al. that highlights strong SEIRA response from hBN nanorod arrays (ref. 41).*

“Whilst in our case we have experimentally demonstrated these concepts using hBN on VO₂, using different combinations of PCMs (such as GeSbTe) and different hyperbolic materials (such as transition metal oxides²⁷), could see applications in a wide frequency range. Ultimately, we anticipate that the combination of low-loss, hyperbolic materials and latchable PCMs could see applications in lithography-free design and fabrication of optical devices.”

- *These last two statements are projected extensions of our current work, on which we are actively working and are demonstrated in the simulation we provide in our manuscript and supplemental information (see Fig. 4, S7 and S8).*

It is seen from the scans that the VO₂ domain boundaries create all possible optical responses.

It is not the boundaries that create all optical responses, it is HPhP propagation at the domains themselves that modifies the polariton wavelength. As such it is the ability to change the permittivity of the PCM domains that is important, the boundaries simply separate the metallic and insulating domains.

If we look at the second out of two Fig. 1, the scans in a) and b) show some patterns only at the corner of the domain, which are quickly disappear then. Thus, it is hard to claim low losses, long propagation distance of the refracted or diffracted polaritons.

Achieving truly low propagation losses is not the primary claim of this paper. In this paper we have shown active control of propagating HPhPs at the nanoscale. Despite this not being the main emphasis of the manuscript, away from the corners of the sample, s-SNOM measurements show propagation ($1/e$) lengths of $2.83\ \mu\text{m}$ (5.2 optical cycles) over dielectric VO_2 , and $0.8\ \mu\text{m}$ (2.5 optical cycles) over metallic VO_2 at an incident frequency of $1450\ \text{cm}^{-1}$ (corresponding to a wavevector of roughly $1.86\ \mu\text{m}^{-1}$ and $3\ \mu\text{m}^{-1}$ respectively). This compares favorably with HPhP propagation within naturally abundant hBN (~ 3.1 and $2.5\ \mu\text{m}$ at the same wavevectors as dielectric/metallic VO_2) and is close to what has been reported for isotopically enriched hBN ($4\ \mu\text{m}$), which is currently the lowest loss Type II hyperbolic material reported to date. Thus, our claims of low losses are in our view appropriate, as they are comparable to the state-of-the-art in hyperbolic systems. Furthermore, it is significantly longer than the state-of-the-art for reconfigurable meta-surfaces (Ref 12), where the propagation length was approximately 2 cycles in the dielectric phase, and no cycles in the metallic phase. In our results, some regions exhibit fast dissipation, but this is because in small corners of the domains, with non-parallel surfaces, multiple HPhPs will all be interfering simultaneously, artificially reducing the propagation lengths from the intrinsic values within the hBN/PCM heterostructure. It is only in large domains where such issues are absent that the intrinsic propagation lengths can be extracted. To clarify this point we have added the following statements to the manuscript:

(page 5 line 12): Propagation lengths ($1/e$) are approximately $2.83\ \mu\text{m}$ (5.2 cycles) and $0.8\ \mu\text{m}$ (2.5 cycles) in the dielectric and metallic phases at this frequency respectively, which is comparable to propagation lengths in naturally abundant hBN (~ 3.1 and $2.5\ \mu\text{m}$ at the same wavevectors respectively)²¹.

It is hard to call this approach as a real control of the surface polariton's propagation.

This conclusion of the Reviewer goes against both our experimental and theoretical data. Our results clearly show that through a temperature-induced phase transition of the PCM from the dielectric to metallic state that the polariton wavelength is significantly modified, observable in all figures. Furthermore, we demonstrate that when measuring the polariton propagation across a PCM phase boundary that the difference in wavelength results in refraction of the polariton. By changing the angle of that phase domain with respect to the polariton initial propagation direction, laws of optics dictate that the refracted and reflected angles can be controlled. We stress in the submitted manuscript (Section 'Towards Refractive Near-Field Optics'), our work illustrates how new device functionalities can be designed through a hyperbolic-PCM heterostructure, and we demonstrate a few specific optical functions that can be reconfigurably designed using this approach (e.g. planar lensing). These are effects that have not been demonstrated previously. As such, we believe we have indeed demonstrated 'real control of the ... polariton's propagation' (the word 'surface' intentionally removed as these modes are not surface confined, as stated above).

To clarify this point we have specifically amended the abstract to make this point clearer from the start of the paper (page 1 line 20)

‘Critically, we show that this system supports in-plane HPhP refraction, thus providing a prototype for a new class of planar refractive optics.’

Finally, the demonstrated realization of the hetero-structure is too far from integrated nanophotonics applications.

We have nowhere claimed that this approach can be immediately implemented in integrated nanophotonic applications. However, one can say the same for just about any current metasurface design at this point, as this field is still in the early stages of development and still even farther from full commercialization. We do note that such a simple heterostructure can be generally applied to the broad and growing class of hyperbolic materials and PCMs, without the need for lithographic fabrication, and thus would indeed be relatively straight-forward to implement. For example in the manuscript (Fig. 4 and Fig. S7 and S8) we simulate structures that could be fabricated using a combination of GeSbTe or VO₂ sputtering, and MOCVD hBN growth.

To address this point we have added the following text to the manuscript (page 8 line 24)

“Although the device we present here is a conceptual prototype, our system could be realized in practice and scaled by using VO₂ or GeSbTe films grown by sputtering, and boron nitride grown by metal organic chemical vapor deposition⁴⁶. There remain numerous material challenges – such as the growth of high quality, large area hBN – in realizing such a system, but this provides a route to achieving scalable reconfigurable devices.”

Summarizing, the manuscript is not suitable for publication in Nature Communications.

In summary, we have demonstrated a reconfigurable, hyperbolic metasurface that differs in function and technical approach from all previous phase-changing and/or polaritonic devices in the literature (see Refs highlighted in Table and below). We also demonstrate a generalized path for realizing such reconfigurable metasurfaces based on a family of HPhP devices using VO₂ or other PCMs such as GST. We therefore believe that the manuscript is eminently suitable for publication in *Nature Communications*.

Reviewer #2 (Remarks to the Author):

The authors demonstrate the control of low-loss hyperbolic phonon polariton in hBN with the help of switching the phase of VO₂ that they use as a substrate. They further demonstrate refraction of the polaritons. Altogether they present a set of experiments that demonstrate the feasibility reconfigurable phonon polariton nanooptics, which could allow, for example, the development of tiny circuits (beam steering, waveguiding, cavities), which could enable novel mid-infrared sensing or signal processing application. The paper is for these reasons very interesting and attractive for the broad readership of Nature Communications.

We thank the reviewer for the positive comments on our work. We address their comments in detail below.

Before publication, the authors should elaborate a bit more some details and address the following points:

1) Numbering of figures needs to be corrected. E.g. there are two Fig. 1.

We apologize for this oversight; the figure labelling was modified upon PDF conversion when submitting the manuscript. We have changed the formatting of the labels so this will not happen on resubmission.

2) Why do we see either (exclusively?) tip- or edge-launched polaritons in Fig. 1b? What determines whether we see tip- or edge-launched polaritons?

There is no exclusive criteria that determines whether a tip- or edge-launched polariton is observed. If the tip and edge happen to be in the excitation laser focus, they both will launch polaritons (the common criteria for both is that sharp structures overcome the momentum mismatch). However, the relative strength of each type of mode is determined by the unique properties of each edge and domain boundary on the sample. Whilst this was briefly discussed in the submitted manuscript, the description was inadequate. We have amended the text to address this point further;

(page 4 line 3) "...derives from the properties of the boundaries in the heterostructure sample, namely the edges of the hBN and VO₂ crystals."

(page 4, line 10) "A more detailed discussion of the occurrence of both tip- and/or edge-launched modes in the s-SNOM images is available in the supplementary information."

We have also added a discussion to the supplementary information, and supplementary Figure S1 (included below), which compares numerical simulations of the edge launched wave for each boundary in our sample. The enhanced electric field strength scattered from the edge of the VO₂ crystal strongly supports our conclusions.

(page 1 supplemental information) "Our sample presents three different interfaces, each of which can have distinctive properties in terms of launching polaritons in the s-SNOM experiment. Observing a tip-launched mode requires a strong reflection from an interface, while observation of an edge-launched mode demands strong scattering off of the sample edge. First, in the present case there is the edge of the hBN flake. The polariton cannot propagate past the edge of the flake and therefore nearly 100% is reflected, leading to a strong tip-launched mode. On the other hand, these hBN flakes are thin (24 nm), and therefore interact only weakly with incident waves, suppressing the edge-launched mode (similar to Ref 25 main text).¹ Thus, we only observe the tip-launched mode near the hBN crystal edge.

Second, there is an interface where the hBN extends over the edge of the VO₂ crystal. As the films of hBN are continuous across the VO₂ edge, tip-launched modes can propagate over this interface and will only be weakly reflected. This has been observed in earlier experiments, for example Ref. 25 of the main text.¹ On the other hand, the VO₂ crystal itself can strongly scatter incident waves to launch polaritons from the VO₂ crystal edges. Therefore, we only see the edge-launched modes at the interfaces between hBN and the VO₂ crystal.

The third type of interface is the domain boundaries between dielectric and metallic VO₂. Due to the relatively small size of the domains in this sample, these show much weaker s-SNOM signals, however, the same arguments as for the edge of the VO₂ crystal hold. Therefore, we mainly see the edge-launched polaritons. This hypothesis is qualitatively supported by electromagnetic simulations of plane waves incident on these three types of boundaries, presented in Fig. S1. The results show that polaritons

launched from the VO₂ crystal edge (Fig. S1a) or dielectric-metal domain boundaries (Fig. S1b) are relatively strong, whilst those initiated from the edge of the hBN flake (Fig. S1c) are relatively weak in intensity.”

Figure S1: Cross sectional plot of electromagnetic fields from hyperbolic polaritons launched at the interface between a) dielectric VO₂ and vacuum, b) dielectric VO₂ and metallic VO₂ and c) at the edge of a hBN flake on VO₂. The peak electromagnetic fields launched above the dielectric domain are $\sim 2.3 \cdot 10^7$ V/m, $\sim 1.8 \cdot 10^7$ V/m and $\sim 1.2 \cdot 10^7$ V/m, indicating the strongest fields are launched at the edge of the VO₂ flake, however, these simulations clearly show that edge-launched modes are highly suppressed in c).

3) In Fig. 1d one can see smaller fringe spacing on the metallic VO₂. How can the authors exclude that we do not see for some reason the tip-launched polaritons? The ratio in wavelength between metallic and dielectric VO₂ is close to a factor 2.

Interpreting polariton wavelengths close to a factor of 2 in difference has to be treated with caution. However, in Fig. S5, our data for 1428 cm⁻¹ excitation (enlarged below for clarity) we see both tip- and edge-launched waves above metallic VO₂, with tip-launched waves exclusively associated with the edge of the hBN, and edge-launched waves exclusively associated with the edge of the VO₂ crystal. As a result we can be confident that in Fig. 1d we are indeed observing an edge-launched wave over metallic VO₂. We have amended the manuscript to address this point;

(page 7, line 10) “In our assignment of the points in Fig. 3a and b, we consider both tip- and edge-launched modes, above both metallic and dielectric VO₂, which can be observed in Fig. S5.”

4) Fig. 1d: The second dark fringe (from the edge where the blue arrow is located) is at the same position on both the dielectric and the metallic VO₂ left of the blue arrow. Why does this metallic VO₂ not modify the polariton wavelength?

This is an interesting observation and we thank the reviewer for noticing it. There could be several reasons for this unusual result. Whilst the light areas in s-SNOM images indicate the boundaries of the metal and dielectric domains when imaging over the VO₂ crystal alone, on the areas with hBN/VO₂ this is no longer necessarily the case. Notably, effects such as hyperlensing (See Ref 35 & 36) could increase the apparent size of the metallic domains in the s-SNOM image when imaging over hBN. Indications of such magnification can be directly observed in our existing s-SNOM images. If you follow a domain that passes over the hBN edge (See new Fig. 1d, and below) we note an apparent discontinuity at the edge of the crystal. This shows that this metallic domain is slightly enlarged over hBN.

Therefore, for the specific wave mentioned by the reviewer, we believe that the polariton has not yet truly started propagating above the metallic domain. This is because the domain is slightly smaller than it appears in the image. Exploring this effect further and demonstrating conclusively that this is the case is beyond the scope of this work, as it would require the comparison of multiple thicknesses of hBN flakes. We admit that we had not previously identified this effect. However, we believe this to be an anomaly, since it is not observed for other domains.

To address this point we have added a comment in the text to this extent, and amended Fig. 1d.

(page 5, line 18) “Note that the VO₂ domains appear to change size when underneath the hBN (as seen by following the black dashed line in Fig. 1d). This arises from hyperlensing by the hBN^{35,36}, which acts to magnify light scattered into a hyperbolic medium, and can give rise to spatial regions on the edge of a domain where the wavelength appears not to change (seen to the left of the blue arrow in Fig. 1d).”

We also added language to this point into the caption:

“In Fig. 1d the black dashed line indicates the edge of the VO₂ metallic domain, extrapolated from the domain outside the hBN crystal.”

4) In the first paragraph of results the authors write that the polaritons launched from sample edges interfere with the tip. I don't understand what is meant. The polariton field at the position of the tip interferes with the incident field at the position of the tip.

Again, we appreciate the reviewer for catching this error. This has been corrected. The modified text now reads:

(page 3, line 22) “Alternatively, polaritons can be directly launched from sample edges and propagate across the surface to interfere with the incident field at the tip, producing fringes with spacing λ_p .^{18,34}”

5) It would be nice if the authors could show that the polariton wavelength at one specific location on the hBN becomes larger (the original one) when switching back from metallic to dielectric VO₂, in order to demonstrate reconfigurable control.

6) The authors refer in main text to Fig. S2 regarding multiple heating/cooling cycles. I don't understand what is shown in Fig. S2 in this regard. It is too little described.

13) The authors describe their work with the word “reconfigurable”. I don't have clear how the metasurface is reconfigurable. This aspect is too little described and not demonstrated. The authors should elaborate this aspect. Particularly, they should describe in more detail how the sample preparation and experiment was done. E.g. was the VO₂ heated before or after the flake was deposited? At which temperature was imaging performed?

Comments 5,6 and 13 relate to the fact that we did not adequately describe or show the process of switching the VO₂ to ‘reconfigure’ the metasurface design.

By reconfigurable, we are referring to the ability to take the hyperbolic-PCM heterostructure and design a given optical element (e.g. a waveguide directing the polariton to a planar lens), then through external stimuli wipe the written element and rewrite to some new arbitrary design (e.g. a waveguide to an optical antenna). In our case, we have demonstrated this effect through multiple heating and cooling cycles, where the arrangement of the VO₂ domains changes on each cycle. To more clearly demonstrate this, we have added a new figure to the supplemental information where we show multiple s-SNOM maps of the same spatial region at the same incident frequency, but collected at various temperatures following successive heating/cooling cycles. Specifically, the heterostructure was heated to the prescribed temperature, measured in s-SNOM, then cooled to room temperature to reset the VO₂ back to the dielectric phase. Whilst in the new Fig. S2 we show two cycles in which we collected s-SNOM data at a range of temperatures, each image in the paper is formed using a similar process of cooling and reheating the same sample. Certain cycles were used to perform multiple consecutive scans with different incident frequencies to probe the refractive effects and explore the polariton dispersion. The polaritonic response in the dielectric and metallic phases were reproducible in all of these experiments, despite changes in the individual domain positions.

From the reviewer’s comments, it is clear that we did not do a sufficient job explaining how these experiments were performed and how we believe this demonstrates the reconfigurable nature of this device design. As such, we have made the following modifications to the main and supplemental text to clarify these results.

(page 4, line 14) “The sample was heated in-situ inside the s-SNOM microscope on a custom-built heating stage.”

(page 4, line 24) “After heating to high temperatures and performing these s-SNOM measurements, allowing the device to cool to room temperature resets the VO₂ crystal to its dielectric state, after which the sample can be reheated to get a different PCM domain pattern (see Fig. S2). This allows us to reconfigure our device to study the propagation of HPhPs in a range of different geometries and at different frequencies within the same device. The large permittivity difference between metallic and

insulating phases of VO₂ therefore presents an excellent platform to manipulate and control polariton propagation within hyperbolic materials.”

We also amended paragraph 1 of ‘Refraction of hyperbolic polaritons’ to read (circa point 6);

(page 5, line 25) “As the positions of domain boundaries induced via thermal cycling of the VO₂ phase change are naturally quasi-random, we implemented multiple heating and cooling cycles (the same process as Fig. S2) to achieve single dielectric-metal interfaces on the VO₂ crystal for study.”

We have also added Fig. S2 and related discussion to the supplemental information;

Figure S2: Thermally induced phase transition in VO₂ for reconfigurable metasurfaces. Here we show a series of *s*-SNOM images taken at the same position as the sample temperature is increased, showing the growth of metallic VO₂ domains, which manipulate polariton propagation in hBN. By cooling the device back to room temperature the device is reset to its dielectric state, and, upon reheating, form a different phase domain pattern.

7) The fringes associated with the refracted polaritons (marked by the black arrows) are not very well seen in the images. I suggest the authors to play with contrast to improve the fringe visibility and eventually show a zoom in to the region of interest. The dashed lines additionally mask the fringes.

While we agree with the reviewer on this point, we have tried multiple different color schemes and contrast levels and find that the figure provided most clearly shows the refracted polaritons. To enhance the observation of these refracted modes, we have highlighted the propagating polaritons with the dashed lines. However, from comparison of the linescans at the refracted and edge-launched polaritons, it can be clearly seen that waves with the same wavelength are propagating at different angles in the same domain and that the refracted angle is not perpendicular to any domain edge, thus clearly shows the phenomena even if the figure contrast cannot be further optimized.

8) I assume the numerical simulation Fig. 2c (respectively Fig. 1 according to Figure caption) does not include the tip. The authors should clarify that.

Yes this is indeed the case and we thank the reviewer for catching this oversight. We have modified the text to account for this as follows:

(page 6, line 14): "In the simulation we excite the structure with plane waves (45° incidence), and at the edges of the VO_2 crystal, polaritonic waves are launched that propagate across the surface, mimicking edge-launched polaritons. Note that we ignore the tip-sample interaction in these simulations. Instead, HPhPs excited at the edge of the VO_2 crystal (blue) propagate in the y -direction within the dielectric phase."

9) The fringes related to the refraction that we see in Fig. 2a,b seem to be all edge launched. Why are there no tip-launched polaritons seen (I see them only on the left hBN edge in Fig. 2b).

As discussed in point 2, as polaritons can propagate across the interface between metallic and dielectric VO_2 , there is insufficient reflection to see a strong tip-launched mode. We note that in principle tip-launched modes should also be refracted, however, this would require a domain boundary close to the hBN crystal edge (where a strong tip-launched mode is observed). To clarify this point we have added the following text:

(page 6 line 27) "We note that whilst in principle these effects should be observable also with a tip-launched waves in s-SNOM images, during our experiments, we were unable to form a VO_2 domain boundary sufficiently close to the flake edge (seen in Fig 1a) to study this effect."

10) In section "Quantifying Polariton Manipulation" the authors mention a pump-laser. I don't understand why and what is pumped. I think the authors simply mean the laser used for imaging. In this case, the word "pump-laser" is misleading.

Again this is indeed the case and we appreciate the reviewer catching this error. The text has been corrected to more accurately describe the measurement.

11) What is the white horizontal line in Figure 2 (respectively 3) and why is it slightly oscillating?

This simply designates a break in the frequency axis in the plot so that the upper and lower Reststrahlen response can be plotted on the same graph with different scales. This allows for the lower Reststrahlen

to be more clearly observed. We have modified the figure caption to clarify this to ensure there is no confusion.

(caption) “The horizontal white line shown in figure (a) and (b) indicates a break in the graph, between upper ($1394\text{-}1650\text{cm}^{-1}$) and lower ($785\text{-}845\text{cm}^{-1}$) Reststrahlen bands”

12) Fig. 3a (respectively 4a) I don't understand well. What does the inset show? The electric field distribution? What does it reveal?

In the reflectance graph one can see three spectra for different width L of the metal. The spectra change as a whole (why?). The authors state that via width L the peaks corresponding to a series of HPhP modes can be tuned (which indeed one can expect), however, the peaks (why so small?) marked by red arrow do not seem to change with L . The authors need to clarify this issue and elaborate the resonances and their tuning more detail.

We apologize for the lack of clarity in our original submission. We did not adequately discuss these simulations or the associated physics. We have made several changes to address this better.

First, we have simplified Fig. 4a, first to show a better schematic illustration of the simulated devices, as opposed to an electromagnetic field profile to make the simulated structure clear. The electromagnetic field profile was removed, as it does not reveal meaningful information about the nature of the resonances in the structure. We have also added reference spectra of the patterned VO_2 domains without hBN, and for hBN on only dielectric VO_2 . This shows that the reflection has been modified quite significantly vs the two reference spectra, even if resonant features are relatively small (as the reviewer pointed out). We highlight that in this work we aim to only show that re-writable resonators are indeed possible using hBN and a PCM – we do not aim to present an optimized structure (which requires optimizing domain size, pitch, VO_2 crystal thickness and hBN thickness). The changes to the main text are laid out below.

(Fig. 4a);

(caption) “Schematic of refractive planar optics and reconfigurable resonators using phase-change materials (a) shows a tunable polariton metasurface of hBN and VO_2 , which consists of a continuous film of hBN 400nm thick on top of 400nm of VO_2 . By patterning the VO_2 with metallic domains we can excite a rewritable reflection profile, which cannot be generated from either of the materials alone. The pitch is 500nm, with a particle length (L).”

(page 8, line 30) "In Fig. 4a we show the numerically calculated reflection spectrum from a hBN crystal on top of a dielectric VO₂ (blue curve), VO₂ patterned with metallic and dielectric domains (red curve), and hBN on top of such a patterned VO₂ structure (black curve). In the simulated spectra for the hBN on top of patterned VO₂, we observe peaks corresponding to a series of HPhP modes. Whilst these peaks are relatively small in amplitude (as this geometry has not been optimized for an intense resonant response), these modes can be tuned in frequency by changing the width and periodicity of the metallic domain (see Fig. S7 and supplemental information for complete discussion). Thus, in principle, by controlling the size and shape of the metallic domain, one can realize a resonant response that previously was only observed in nanofabricated structures of hBN^{20,39-41}. This implies that such resonators can be realized without the additional induced losses that are incurred with most nanofabrication approaches⁴²⁻⁴⁴. Such resonators could also be achieved experimentally by adding titanium to change the local phase transition properties of VO₂⁴⁵."

Second, we have added a more comprehensive discussion of the tuning of the reflectance spectra to the supplementary information, discussing the effect of the size and pitch of the metallic domain. We specifically highlight that the spectra change as a whole when we simply change the size of the metallic domain, even with no hBN present. This is why the spectrum changed as a whole in the originally submitted manuscript. We clarify that while the resonances presented in the original submitted manuscript tune only a small amount, more significant tuning can be achieved by varying the pitch of the nanostructure (for constant fill fraction). We then explain the origin of this tuning behavior; this structure behaves more similarly to a grating coupler than a localized polariton resonator.

(Page 7 Supplemental Information) "In Fig. 4a of the main text, we present the simulated reflectance spectrum of a hBN film on top of VO₂ patterned into metallic and dielectric domains (hereafter referred to as VO₂ resonators). Here we address the frequency tuning of the resonances observed in these spectra. We address two approaches to achieving frequency tuning – changing resonator size L (L=250 nm in Fig. 4a) with fixed pitch P (P=500 nm in Fig. 4a) and changing pitch with fixed filling fraction ($f = L/P$). The reflectance spectrum of such reconfigurable resonators as a function of the resonator size, is provided in Fig. S7a and illustrates both a variation in reflectance, and small changes in the spectral positions of the resonant modes. The change in the overall reflectance can be attributed largely to changes in the reflectance of the VO₂ resonators, with larger resonators exhibiting higher reflection. The spectral mode shifts are approximately 3.34, 3.34, 4 and 2 cm⁻¹ from lowest to highest modal wavenumbers, which is much lower than what would be expected for localized resonances. The reason for this becomes clear when we consider the influence of grating pitch with a constant fill fraction in Fig. S7b. Here we see that each mode red-shifts significantly (approximately 30 cm⁻¹ for the mode around 1525 cm⁻¹) with increasing grating pitch. Note that here the overall reflectance does not change significantly, as the fraction of metallic vs dielectric VO₂ remains approximately constant when we fix the fill fraction. Our resonant tuning behaviour is consistent with our metasurface effectively acting as a grating coupler, as HPhPs can freely propagate through the hBN film. We note that while here we do not show significant absorption or reflection resonances induced by the metasurface design, this could

potentially be achieved by optimizing the combination of hBN thickness, VO₂ crystal thickness and lateral size, along with the designed metallic domain pitch and size.”

Figure S3: Frequency tuning of hyperbolic modes for the metasurface presented in Fig 4a. a) Shows tuning of the resonant modes for different metallic domain sizes at constant pitch, b) the tuning of the resonant modes for variable domain pitch.

14) Generally, the authors could improve a bit the presentation of results: presentation of figures (labeling, font sizes (e.g. in S3) etc), description of figures in more detail, etc.

We thank the reviewer for the comment. As detailed above, it was clear that this is the case and we have made significant changes to the manuscript to address these issues. We have highlighted many of these above, including the corrections requested to the original Fig. S3 (new S5) and the clarity of the descriptions provided in the figure captions.

In closing, we would like to thank you and both reviewer’s for taking the time to consider our manuscript and believe that by addressing the issues they have raised, that our manuscript is significantly improved. Thank you again for the opportunity to address these comments and we look forward to hearing your decision.

Comparisons to prior works cited

Ref. 8: *Control of plasmonic nanoantennas by reversible metal-insulator transition* – this work details the impact of PCMs upon the localized surface plasmon resonant modes within gold nanoantennas.

Ref 12: *Reversible optical switching of highly confined phonon-polaritons with an ultrathin phase-change material* – this work highlights the use of a PCM (here GST) to modify the surface polariton response of a polar dielectric. They demonstrate the ability to direct write confined regions where a phase change is induced and demonstrate that these regions can be used to launch polaritons in the underlying polar crystal. The authors demonstrate the reversibility of this approach and in our view this is one of the more influential works in active nanophotonics. However, due to the surface confined nature of surface phonon polaritons and the high absorption in the metallic GST domains, polaritons could not be supported within these metallic regions. Therefore, polaritons could only be propagated over one phase of the PCM and thus full optical control via transmission and refraction of the polaritons could not be observed, as is the case in our work.

Ref. 13: *Beam switching and bifocal zoom lensing using active plasmonic metasurfaces* – This work illustrated the use of GST as an active component for beam steering with circular polarization selectivity. The device consisted of a gold metasurface with sub-diffractive elements to directly control the polarization and beam path upon changing the PCM.

Ref. 14: *All-dielectric phase-change reconfigurable metasurface* – This work highlights a chalcogenide-based all-dielectric metasurface that does not incorporate polaritonic effects, and uses GST to induce tunability. By tuning between the dielectric and metallic states, the absorption spectra of the metasurface can be tuned.

Ref. 15: *Conformal coating of a phase change material on ordered plasmonic nanorod arrays for broadband optical switching* – In this paper the authors discuss conformal coatings of VO₂ on vertically aligned ITO nanorod arrays. They demonstrate that the phase change induced can tune the absorption spectra by the influence of the change in the local index of refraction upon the polariton modes.

Ref. 16: *Ultra-thin perfect absorber employing a tunable phase change material* – In this work the authors detail the use of VO₂ as a highly lossy dielectric, which when deposited at ultrathin layers can work as a perfect absorbing film. This builds on the author's prior work with ultra-thin Ge films.

Ref. 17: *Tunable hyperbolic metamaterials utilizing phase change heterostructures* – Here the authors fabricate a layered metamaterial using sub-wavelength layers of VO₂ and TiO₂. By transitioning from the dielectric to the metallic state of VO₂, the optical response of this metamaterial transitions from an elliptical polaritonic dispersion to a hyperbolic response within the near-IR. The optical losses within the metallic regime limit the observation of hyperbolic polariton propagation. Furthermore, this work does not demonstrate reconfigurable HPP modes, instead demonstrate modulation of the behavior from SPP to HPP.

Ref. 25: *Launching Phonon Polaritons by Natural Boron Nitride Wrinkles with Modifiable Dispersion by Dielectric Environments* – This paper highlights some of the interesting effects in hyperbolic polariton launching when the reflection at a boundary (here a wrinkle) is less than 100%, resulting in changes in the tip- and edge-launched constituents of the modes. They further demonstrate that the polariton

dispersion is modified by comparing the propagation in hBN over SiO₂ and Au. This does not discuss polariton refraction or incorporate phase change materials.

Ref. 29: *Manipulation and steering of hyperbolic surface polaritons in hexagonal boron nitride* – This work highlights the impact of the hBN structural shape in dictating the polariton propagation and the standing waves that result. This is clearly distinct from our efforts.

Ref. 30: *Infrared hyperbolic metasurface based on nanostructured van der Waals materials* – Here the authors demonstrated an in-plane hyperbolicity within hBN by patterning the flake into stripes using a focused ion beam, similar to a grating structure. While demonstrating a hyperbolic metasurface, this again is not reconfigurable as once the structure is fabricated it is fixed. Our work builds upon these results (published earlier this Spring in Science), by demonstrating that through the incorporation of a PCM, that one could design metasurfaces such as this one, then reconfigure this surface into a different design.

Reviewers' Comments:

Reviewer #1:

Remarks to the Author:

The authors have provided detail response but have not addressed my concerns. In some sense the authors just repeated and reworded their arguments presented in the manuscript. In my opinion the table in the authors' response clearly confirms my previous conclusion that the manuscript reports just minor development in the field and is not suitable for NC publication.

Reviewer #2:

Remarks to the Author:

The authors provide detailed answers and corrected the manuscript accordingly. In my opinion it can be published now.